# Betulinic Acid Improves Cardiac-Renal Dysfunction Caused by Hypertrophy through Calcineurin-NFATc3 Signaling

**DOI:** 10.3390/nu13103484

**Published:** 2021-09-30

**Authors:** Mi-Hyeon Hong, Se-Won Na, Youn-Jae Jang, Jung-Joo Yoon, Yun-Jung Lee, Ho-Sub Lee, Hye-Yoom Kim, Dae-Gill Kang

**Affiliations:** 1Hanbang Cardio-Renal Research Center, Wonkwang University, Iksan 54538, Korea; m-hyeon123@naver.com (M.-H.H.); sewon3066@naver.com (S.-W.N.); j8626@naver.com (Y.-J.J.); mora16@naver.com (J.-J.Y.); shrons@wku.ac.kr (Y.-J.L.); host@wku.ac.kr (H.-S.L.); 2College of Oriental Medicine and Professional Graduate School of Oriental Medicine, Wonkwang University, Iksan 54538, Korea

**Keywords:** betulinic acid, cardiac hypertrophy, isoprenaline, calcineurin-NFATc3 signaling

## Abstract

Cardiac hypertrophy can lead to congestive heart failure and is a leading cause of morbidity and mortality worldwide. In recent years, it has been essential to find the treatment and prevention of cardiac hypertrophy. Betulinic acid (BA), the main active ingredient in many natural products, is known to have various physiological effects. However, as the potential effect of BA on cardiac hypertrophy and consequent renal dysfunction is unknown, we investigated the effect of BA on isoprenaline (ISO)-induced cardiac hypertrophy and related signaling. ISO was known to induce left ventricular hypertrophy by stimulating the β2-adrenergic receptor (β_2_AR). ISO was injected into Sprague Dawley rats (SD rats) by intraperitoneal injection once a day for 28 days to induce cardiac hypertrophy. From the 14th day onwards, the BA (10 or 30 mg/kg/day) and propranolol (10 mg/kg/day) were administered orally. The study was conducted in a total of 5 groups, as follows: C, control; Is, ISO (10 mg/kg/day); Pr, positive-control, ISO + propranolol (10 mg/kg/day); Bl, ISO + BA (10 mg/kg/day); Bh, ISO + BA (30 mg/kg/day). As a result, the total cardiac tissue and left ventricular tissue weights of the ISO group increased compared to the control group and were significantly reduced by BA treatment. In addition, as a result of echocardiography, the effect of BA on improving cardiac function, deteriorated by ISO, was confirmed. Cardiac hypertrophy biomarkers such as β-MHC, ANP, BNP, LDH, and CK-MB, which were increased by ISO, were significantly decreased by BA treatment. Also, the cardiac function improvement effect of BA was confirmed to improve cardiac function by inhibiting calcineurin/NFATc3 signaling. Renal dysfunction is a typical complication caused by cardiac hypertrophy. Therefore, the study of renal function indicators, creatinine clearance (Ccr) and osmolality (BUN) was aggravated by ISO treatment but was significantly restored by BA treatment. Therefore, it is thought that BA in cardiac hypertrophy can be used as valuable data to develop as a functional material effective in improving cardiac-renal dysfunction.

## 1. Introduction

Heart failure is a risk factor of many cardiovascular diseases and is considered globally as one of the causes of morbidity and mortality [1]. Cardiac hypertrophy is a leading cause of heart failure and is induced by pathological stimuli such as myocardial infarction, cardiac fibrosis, inflammation, and ventricular remodeling [2]. Cardiac hypertrophy is induced by biomechanical stress and characterized by thickening of the ventricle wall in the heart, the growth of heart mass and cardiomyocytes [3]. Chronic cardiac hypertrophy caused apoptosis of cardiomyocytes and would cause heart failure and sudden death. Cardiac hypertrophy is a crucial reason for heart failure; therefore, the preparation of therapeutic solutions and prevention of cardiac hypertrophy is important. 

β-adrenergic signaling, which was activated by sympathetic stimuli, plays an important role in progression of cardiac hypertrophy [4]. Chronic β-adrenergic receptors (βAR) are located in cardiomyocytes, fibroblasts and endothelial cells. βAR stimulation induces an increase in the size of cardiomyocytes. ISO is an agonist of βAR and is used to induce cardiac hypertrophy in animal models [5]. When increasing intracellular calcium concentration, calcineurin is activated. The activating calcineurin binds to the nuclear factor of activated T-cells (NFAT) and the NFAT was translocated to the nucleus. The NFAT induced activation of the hypertrophic genes, such as atrial natriuretic peptide (ANP), brain natriuretic peptide (BNP), and beta myosin heavy chain (β-MHC). Calcineurin-NFAT signaling is important in regulating cardiac hypertrophy [6]. Propranolol is a non-selective β blocker and has been used to treat cardiac vascular disease such as hypertension and cardiac arrhythmias by reducing heart rate and blood pressure. Also, propranolol lowered heart weight and the thickness of the left ventricular wall in mice and is effective in cardiac function [7,8].

In recent years, research on traditional herbal medicine has increased, and the potential value of herbs rediscovered [9]. Betulinic acid (BA) is an active compound in the bark of birch tree *Betula* spp. (Betulaceae). BA has various biological activity such as anti-inflammatory, anti-viral, anti-oxidation and anti-cancer [10]. BA suppresses fibrosis in diabetic rat kidneys and inhibits cardiomyocyte apoptosis [11]. However, the effect of BA on ISO-induced left ventricle hypertrophy and detailed signaling were not investigated. 

Therefore, the aim of present study was to determine the effect of BA in left ventricle hypertrophy induced by ISO and to clarify its underlying signaling. 

## 2. Materials and Methods

### 2.1. Chemicals and Meterials

Betulinic acid (BA) was purchased from Toronto Research Chemicals (Toronto, Canada). Isoprenaline (ISO) was purchased from Sigma Chemical Corp. (St Louis, MO, USA). Propranolol hydrochloride was bought from Sigma Chemical Corp. (St Louis, MO, USA) and the catalog number is 4199-10-4. MYH7, NFATc3, p-NFATc3, GAPDH were purchased from Santa Cruz Biotechnology (Dallas, TX, USA). ANP and BNP were purchased from Abcam (Cambridge, UK). Calcineurin was purchased from BD Bioscience in San Jose, CA, USA. Secondary antibody, such as rabbit and mouse, was purchased from ENZO Life Sciences (Farmingdale, New York, NY, USA). 

### 2.2. Animals and Treatment

Male Sprague Dawley rats (SD rat), 6 weeks old, were purchased from Koatech (Pyeongtaek, Korea), and allowed free access to food and water under the 12 h light and dark cycle. All animal experiments were conducted within the guidelines of Wonkwang University. After the period of adaptation for 7 days, the rats were randomly divided into 5 groups: C, control; Is, ISO (10 mg/kg/day); Pr, positive-control, ISO + propranolol (10 mg/kg/day); Bl, ISO + BA (10 mg/kg/day); Bh, ISO + BA (30 mg/kg/day). Each group was composed of 8 rats. The ISO induced left ventricular hypertrophy by intraperitoneal injection for 4 weeks at a dose 10 mg/kg/day daily. PRO 10 mg/kg/day, BA 10, 30 mg/kg/day each was fed with orally intake. The experimental animals were approved by the Institutional Animal Care and Use Committee of the Wonkwang University (WKU20-115).

### 2.3. Echocardiographic Assessments

Cardiac function was assessed by using echocardiography, which was carried out using an ultrasound imaging system with 18LS probe at a frequency of 14 MHz (VINNO6, Vinno Corpration, China). Rats were anesthetized with 4% isoflurane inhalation mixed with oxygen, before shaving and experiment. Rats were measured, then fixed with the nose cone function, which maintained anesthesia by 1.5% isoflurane at a flow rate of 1 L/min. The parameters, such as fractional shortening (FS), ejection fraction (EF), left ventricle intraventricular diameters at systole (LVIDs), left ventricle intraventricular diameters at diastole (LVIDd), were measured by VINNO LV function software and recorded 2D-guided M mode cross section. 

### 2.4. Histological Analysis

Heart tissue was extracted from 3 rats and fixed in 4% formaldehyde. Heart tissues were embedding in paraffin and were cut in 6 μm thick sections. The tissues were stained with hematoxylin for 4 min and eosin for 2 min. Picrosirius Red stain Kit was performed by instructions (Picrosirius Red stain, Polysciences, Warrington, PA, USA). The paraffin embedded cardiac tissue was deparaffinized and hydrated with water. The tissue was stained with hematoxylin for 8 min and treated with solution A, B, C, equipped with staining kit. Hematoxylin and Eosin staining and picrosirius red staining were examined by Motic Easy Scan One (Meyer Instruments, INC., Houston, TX, USA). To confirm cardiac-tissue sizes, wheat germ agglutinin (WGA) staining was performed. The slide of cardiac tissue was fixed in 4% paraformaldehyde for 15 min and washed 3 times. The 5 μM WGA was used for staining cell surface and diluted with 1xPBS. The tissue was treated with 5 μM of WGA solution for 20 min. The stained slides were determined by EVOSTM M5000 microscopy (Thermo Fisher scientific, Waltham, MA, USA). Whole heart tissue was observed by EVOS M5000 microscopy and edited by Celleste Image Analysis Software (Celleste 5, Thermo Fisher Scientific, Waltham, MA, USA) program for histological analysis.

### 2.5. Immunoblotting

Protein in the left ventricle was extracted by PRO-PREP^TM^ protein extraction solution (Seongnam, Republic of Korea). Left ventricle tissue was resolved by 10% SDS polyacrylamide gel electrophoresis (PAGE) and transferred nitrocellulose membranes. The membranes were blocked 4% BSA solution with 1xTBS for 2 h and incubated with primary antibody overnight. The membranes were washed 3 times for 10 min each and conjugated with a second antibody for 2 h. The bands were detected with chemiluminescent substrate (WestGlowTM FEMTO, BIOMAX, Seoul, Republic of Korea). The first antibodies used in the experiment, including NFATc3, p-NFATc3, GAPDH were diluted in 1:1000 with 4% BSA. ANP and BNP were diluted 7 μL with 10 mL of 4% BSA. Rabbit and mouse, which were used as a secondary antibody in western blot, were diluted in 1:5000 with TBS-T. Left ventricular tissue from each group was used for Western blot (*n* = 5 for each group).

### 2.6. Measurements of Biomarkers in Serum

Blood samples were contained in the vacutainer with heparin and centrifuged at 3000 rpm for 20 min at 4 °C. Serum was separated in centrifuged blood samples. Biomarkers of serum in cardiac hypertrophy, including CK-MB, ALT, AST, LDH, albumin, creatinine, were measured by a clinical chemistry analyzer, which was FUJI DRICHEM NX700 (Tokyo, Japan). To measure LDH level, the serum had to be diluted with distilled water; therefore, distilled water had to be installed in the D2 of the FUJI analyzer. The serum samples were equipped and measured by QC card. 

### 2.7. Monitoring of Renal Function

Rats in each group were isolated from metabolic cages for 4 days (*n* = 5 for each group), and urine samples from each rat were collected for 24 h. The urine samples were centrifuged at 13,000 rpm, for 20 min and stored at 4 °C. The collected urine samples were used to measure creatinine, osmolality, albumin, and creatinine clearance. The creatinine level in the urine was measured by a colorimetric method using a spectrophotometer (New York, NY, USA). Osmolality was measured by using an Advanced CRYOMATICTM osmometer Model 3900 (Norwood, MS, USA).

### 2.8. Statistical Analysis

All data were repeated at least three times and statistical analysis was performed by using Student’s *t*-test. *p* < 0.05 was considered statistically significant.

## 3. Results

### 3.1. BA Ameliorated Heart Function in ISO-Induced LV Hypertrophy Rat

Heart weight (HW) ratio to body weight (BW) increased in ISO treatment. LV weight also increased in ISO treatment; however, total HW ratio and LV weight ratio to BW decreased significantly in the BA treatment groups (Figure 1A,B). Echocardiography was performed to measure heart function. EF, FS was significantly decreased in the ISO group. However, the EF, FS level was increased in BA treatment groups (Figure 2B,C). Stroke volume (SV) was decreased in the ISO group. The SV increased in PRO 10, Bl and Bh treatment groups (Table 1). LVd Mass also was declined in the ISO group; however, the LVd mass was restored in the Bh group (Table 1). The percentage of LVPW was lowered in the ISO group compared with the control group. The percentage of LVPW was elevated in the PRO 10 and Bl groups (Table 1). IVSd, LVIDd, LVPWd and LVPWs dropped in the ISO group; however, BA restored the levels markedly, (Table 1).

### 3.2. BA Changed Level of Serum Biomarkers in ISO-induced LV Hypertrophy Rat

Table 2 showed serum biomarkers such as LDH, AST, ALT, CK-MB, BUN, albumin, creatinine. The LDH, CK-MB, and ALT were significantly activated in the ISO group compared with the control group. In treatment BA groups, activation of LDH, CK-MB, and ALT was less than in the ISO group. There was no significant change in the AST level in each group. BUN, which is a kidney function marker, was increased in the ISO group and the level reduced in the Bh group. The level of albumin in serum was lowered in the ISO group compared with the control group; however, the level was restored in the Bh group (Table 2). Creatinine in serum was not different in each group (Table 2).

### 3.3. BA Has Influenced Heart Morphology 

Hematoxylin and Eosin staining was performed to verify the size change in cardiac tissues by BA treatment in ISO-induced cardiac hypertrophy. Hematoxylin and Eosin staining showed that ISO induced cardiac hypertrophy compared with the control. Whole heart tissue was smaller in BA treatment groups than in the ISO groups (Figure 3). WGA was stained with cell surface, therefore, the cell size was checked. Cell size in the heart tissue was increased in the ISO treatment group compared with the control group. BA decreased the cell size of the heart (Figure 3). To evaluate the protective effect on fibrosis of the heart tissue, picrosirius red staining was performed. Fibrosis was found red in picrosirius red staining of tissue. ISO also caused fibrosis in the heart tissue and the fibrosis area induced by ISO was found in picrosirius red staining (Figure 3). However, BA ameliorated the fibrosis in the heart tissue (Figure 3).

### 3.4. BA Decreased Cardiac Hypertrophy Marker in ISO-induced LV Hypertrophy Rat

β-MHC is one of markers in cardiac hypertrophy and was increased in the ISO treatment group. The expression was decreased in the PRO 10, BA 10 and BA 30 groups (Figure 4A). ANP and BNP expression were increased in the ISO group; however, the expression declined in Bh group (Figure 4A).

### 3.5. BA Attenuated Kidney Function in ISO-induced LV Hypertrophy Rat

Left kidney weight was lowered in the ISO group, but the weight was remarkably increased in the BA group (Figure 5A). Right kidney weight was not changed in each group. Osmolality, urine albumin, urine creatinine and BUN were measured to identify the BA effect. The markers of kidney function, such as osmolality, urine albumin, urine creatinine and BUN were deteriorated in the ISO group (Figure 5B–E). The markers were improved in the BA treatment groups (Figure 5B–E). Creatinine clearance (Ccr) level was decreased significantly in the ISO group; however, the Ccr was enhanced in the Bh group (Figure 5F).

### 3.6. BA Inhibited Calcineurin/NFATc3 Signaling

Isoprenaline promoted calcineurin and phosphorylation NFATc3 expression level and induced left ventricle hypertrophy in rats. However, Bh suppressed the activation calcineurin and p-NFATc3 in left ventricle tissue (Figure 4B).

## 4. Discussion

Our results show that BA influenced ISO-induced cardiac hypertrophy and kidney function. Treatment with BA reduced heart weight and left ventricle weight in ISO-induced cardiac hypertrophy, and also improved heart function and cardiac fibrosis. ISO was used in our study because it is known to induce left ventricular hypertrophy and chronic stimulation at β-adrenergic receptors, leading to cardiac hypertrophy and cardiac fibrosis [12]. Cardiac hypertrophy markers, such as ANP, BNP and β-MHC were decreased in the BA treatment group, especially in the high-concentration BA group. Also, BA restored the serum biomarkers, such as LDH, CK-MB, ALT, AST, albumin and creatinine in ISO-induced LV hypertrophy. BA decreased calcineurin and p-NFATc3 activation in ISO-induced cardiac hypertrophy. To induce LV hypertrophy, ISO was used in the present study. ISO caused to LV hypertrophy and chronic stimulation in β-adrenergic receptor by injecting isoprenaline induced cardiac fibrosis [12]. In Figure 3, picrosirius red staining showed that BA attenuated cardiac fibrosis in ISO-induced cardiac hypertrophy. While treating isoprenaline, the left ventricle wall and weight increased. The BA treatment group decreased whole heart, left ventricle weight and attenuated heart fibrosis. Isoprenaline treatment damaged cardiac function [13]. The group of rats with cardiac hypertrophy induced by isoprenaline deteriorated the figure of SV, %LVPW, IVSd, LVPWd, LVPWs compared with the control group in echocardiography. The BA treatment group, especially BA 30 high-treatment group, attenuated the level of echocardiography.

Chronic injection of ISO into rats changed the biomarker in serum, for example, LDH, CK-MB, ALT, AST, albumin, and creatinine [14]. Myocardial injury by ISO altered the level of LDH, CK-MB, ALT and AST in serum [15]. The levels of biomarkers in serum were elevated in ISO-induced cardiac hypertrophy rat; however, the level restored in BA 10 and 30 groups. The ISO activated β-adrenergic signaling. β-adrenergic receptor regulates Ca^2+^ concentration and Ca^2+^ alteration activated calcineurin/NFATc3 signaling [16,17]. The activated β-adrenergic signaling by ISO injection led to LV hypertrophy through calcineurin/NFATc3 pathway. Calcineurin was characterized as an important regulator in heart development and pathological heart remodeling [18,19]. NFATc3, which is one of NFAT proteins family, is activated by phosphorylation of calcineurin. NFATc3 has a regulatory domain that has dephosphorylated the NFATc3, as a result of activation of calcineurin in the T-cell. The dephosphorylation of NFATc3 is translocated to the nucleus and combined with GATA 4 and BNP gene [20]. Activated calcineurin/NFATc3 signaling was established to transduce cardiac hypertrophy signaling in cardiomyocytes. ISO-induced cardiac hypertrophy in rats increased the expression of calcineurin and phosphorylated NFATc3 in LV tissue. In these results, Bh suppressed dephosphorylated NFATc3. It is possible to measure the activity of NFATc3 in the nucleus, but it could not be confirmed in this study, and further research is needed. The results of phosphorylation NFATc3 expression in animals are also meaningful [21]. The NFAT activation caused cardiac hypertrophy markers, which were β-MHC, ANP and BNP [22,23]. ISO increased the expression of β-MHC, ANP and BNP in left ventricle tissue. The result showed that BA altered the level of β-MHC, ANP and BNP in ISO-induced cardiac hypertrophy in rats. Chronic cardiac hypertrophy has affected to kidney function. Cardiac hypertrophy, such as left ventricular hypertrophy is crucial one triggered kidney dysfunction and fibrosis [24,25]. Osmolality level in urine is established as indicator of kidney function and altered in chronic kidney disease patients [26]. Serum creatinine, BUN, serum albumin, urinary kidney injury molecule 1 (KIM-1) et al were major predictor for kidney function and measured to diagnose the kidney [27,28]. Osmolality, albumin, creatinine level in urine got worse in cardiac hypertrophy induced by ISO compared with control group. The biomarkers for kidney function were attenuated significantly in BA treatment groups. Creatinine secretion in blood and urine was triggered by renal filtration and tubular secretion. When chronic kidney failure occurs, there is a problem with the secretion of creatinine. Ccr is calculated by creatinine concentration in urine and blood; therefore, Ccr is an important indicator of renal function [29,30]. Ccr level was degenerated in the ISO group compared with the control group. Bh improved the Ccr level significantly. However, the creatinine level in serum is also an important factor in kidney function and has been related with Ccr calculation. In this study, the creatinine value in serum was changed in each group, but there was no significance. Although serum creatinine was not significant, the biomarkers for kidney function such as Ccr, osmolality, and BUN were altered by BA treatment. Thus, the BA is suggested as a therapeutic solution for cardiac hypertrophy and has effect on kidney function. 

However, major limitations of this study are acknowledged and need to be addressed. The effects of BA in chronic cardio-renal dysfunction are not clear. A new animal model for chronic cardio-renal syndrome should be established and there is a need for a new experiment on the effects of BA on chronic cardio-renal dysfunction. 

## 5. Conclusions

Our study results confirmed that BA treatment significantly improved heart weight, cardiac fibrosis, cardiac function, and renal function in the ISO-induced cardiac hyper-trophy model. This improvement was confirmed, showing an improvement effect by inhibiting calcineurin/p-NFATc3 signaling. Therefore, it is thought that this research on the effects of BA in cardiac hypertrophy can be used as valuable data to develop as functional material effective in improving cardiac-renal dysfunction (Figure 6).

## Figures and Tables

**Figure 1 nutrients-13-03484-f001:**
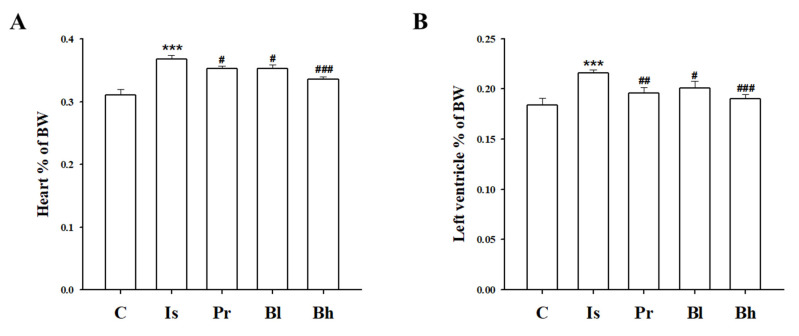
Effect of BA on whole heart and left ventricular tissue weight. The ratio heart (**A**) and left ventricle weight of body weight (**B**) were altered in Bl and Bh groups. C, control; Is, hypertrophy rat, isoprenaline (10 mg/kg/day) intraperitoneal injection group; Pr, positive control, Is + propranol (10 mg/kg/day); Bh, Is + betulinic acid (30 mg/kg/day). Data were expressed as the means ± SD (*n* = 8 for each group). *** *p* < 0.001 vs. C, ^#^
*p* < 0.05, ^##^
*p* < 0.01, ^###^
*p* < 0.001 vs. Is.

**Figure 2 nutrients-13-03484-f002:**
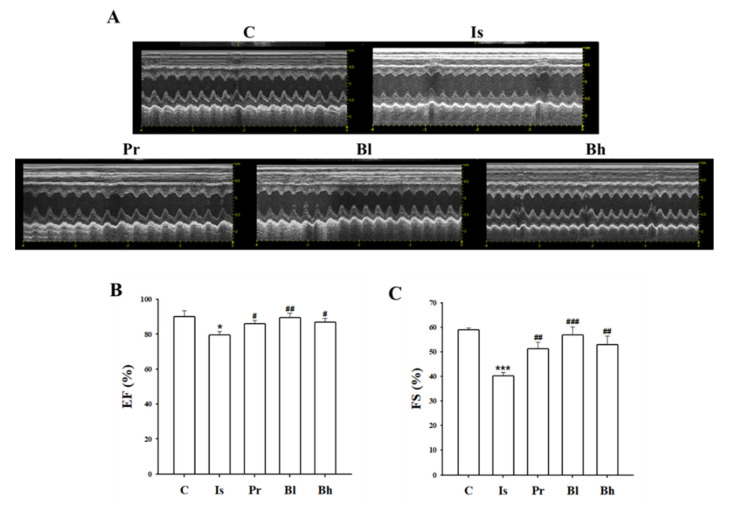
Effect of BA on cardiac function. Representative images of echocardiography of each group (**A**). EF (**B**) and FS (**C**) ratio deteriorated in Is compared with C, however, the data significantly improved in BA treatment groups. C, control; Is, hypertrophy rat, isoprenaline (10 mg/kg/day) intraperitoneal injection group; Pr, positive control, Is + propranol (10 mg/kg/day); Bh, Is + betulinic acid (30 mg/kg/day); EF, ejection fraction; FS, fractional shortening. Data were expressed as the means ± SD (*n* = 5 for each group). * *p* < 0.05, *** *p* < 0.001 vs. C, ^#^
*p* < 0.05, ^##^
*p* < 0.01, ^###^
*p* < 0.001 vs. Is.

**Figure 3 nutrients-13-03484-f003:**
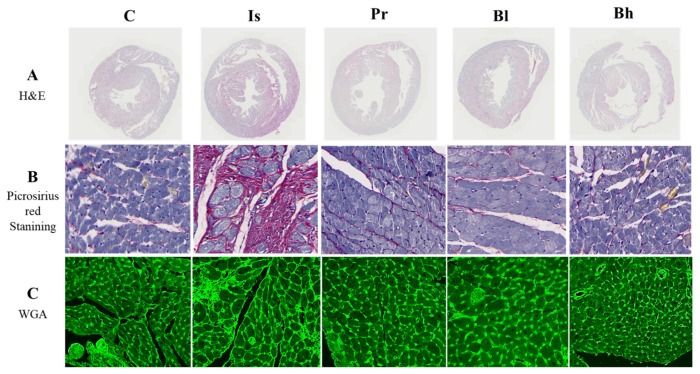
Effect of BA on morphology and fibrosis in heart. Representative images comparing cardiac hypertrophy (**A**, hematoxylin-eosin staining staining) and cardiac fibrosis (**B**, picrosirius staining). Representative WGA staining of the heart sections (**C**). Green color represents cardiomyocyte boundary a typical cardiomyocyte size (magnification, ×400). The overall heart size and myocardial fibrosis were increased in the Is group compared to the C group, and the improvement effect was confirmed in the Pr and Bh groups.C, control; Is, hypertrophy rat, isoprenaline (10 mg/kg/day) intraperitoneal injection group; Pr, positive control, Is + propranol (10 mg/kg/day); Bh, Is + betulinic acid (30 mg/kg/day); WGA, wheat germ agglutinin.

**Figure 4 nutrients-13-03484-f004:**
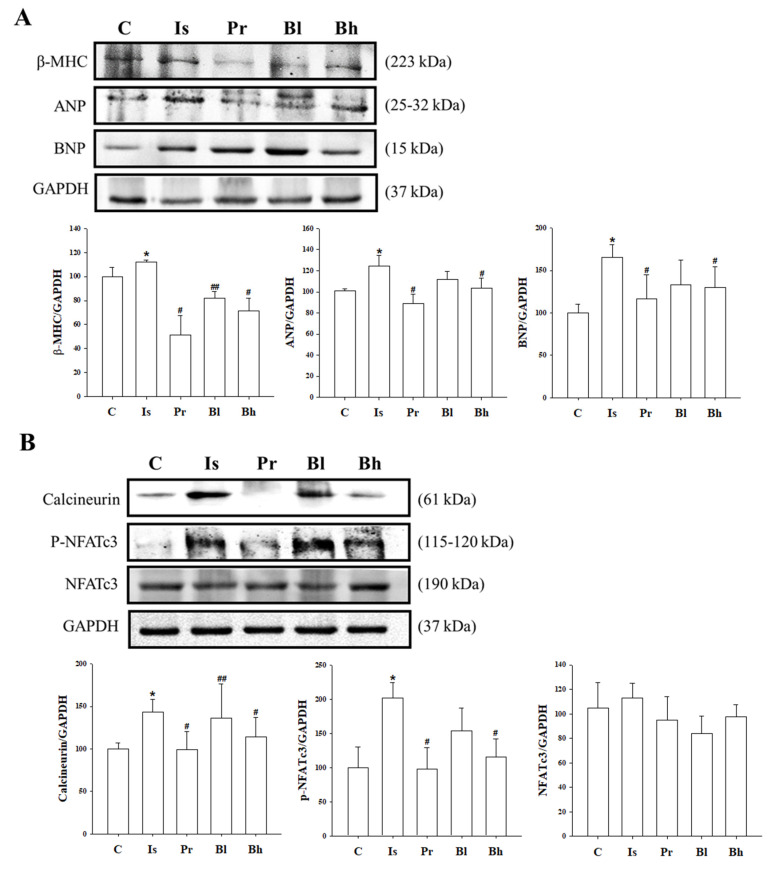
Effect of BA on cardiac hypertrophic biomarkers and calcineurin/NFATc3 signaling. (**A**) β-MHC, ANP, BNP protein level altered by BA treatment in western blotting. (**B**) The calcineurin/NFATc3 protein activation was inhibited in Bh group. C, control; Is, hypertrophy rat, isoprenaline (10 mg/kg/day) intraperitoneal injection group; Pr, positive control, Is + propranol (10 mg/kg/day); Bh, Is + betulinic acid (30 mg/kg/day); NFAT, nuclear factor of activated T cells. Data were expressed as the means ± SD (*n* = 5 for each group). * *p* < 0.05 vs. C, ^#^
*p* < 0.05, ^##^
*p* < 0.01 vs. Is.

**Figure 5 nutrients-13-03484-f005:**
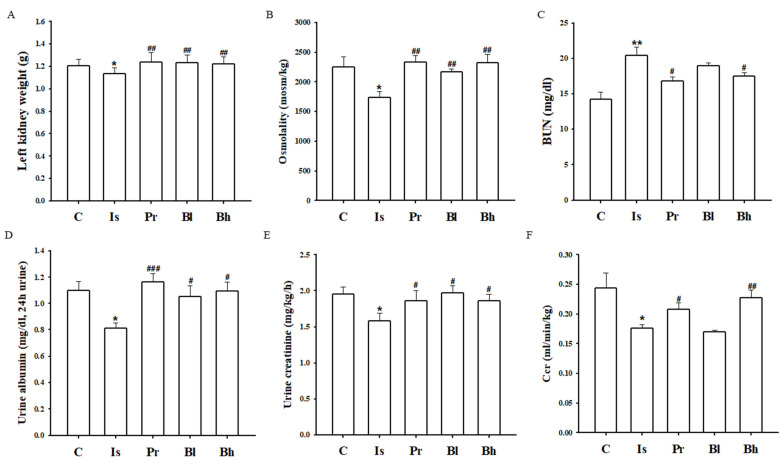
Effect of BA on kidney function. (**A**) The left kidney weight was assessed in each group. (**B**) The osmolality was measured by using urine. (**C**) BUN concentration was measured in each group. (**D**) Albumin concentration in urine was checked in each group. (**E**) Urine creatinine level showed in each group. (**F**) Ccr was calculated in each group. C, control; Is, hypertrophy rat, isoprenaline (10 mg/kg/day) intraperitoneal injection group; Pr, positive control, Is + propranol (10 mg/kg/day); Bh, Is + betulinic acid (30 mg/kg/day); BUN, blood urea nitrogen; Ccr, creatinine clearance. Data were expressed as the means ± SD (*n* = 5 for each group). * *p* < 0.05, ** *p* < 0.01 vs. C, ^#^
*p* < 0.05, ^##^
*p* < 0.01, ^###^
*p* < 0.001 vs. Is.

**Figure 6 nutrients-13-03484-f006:**
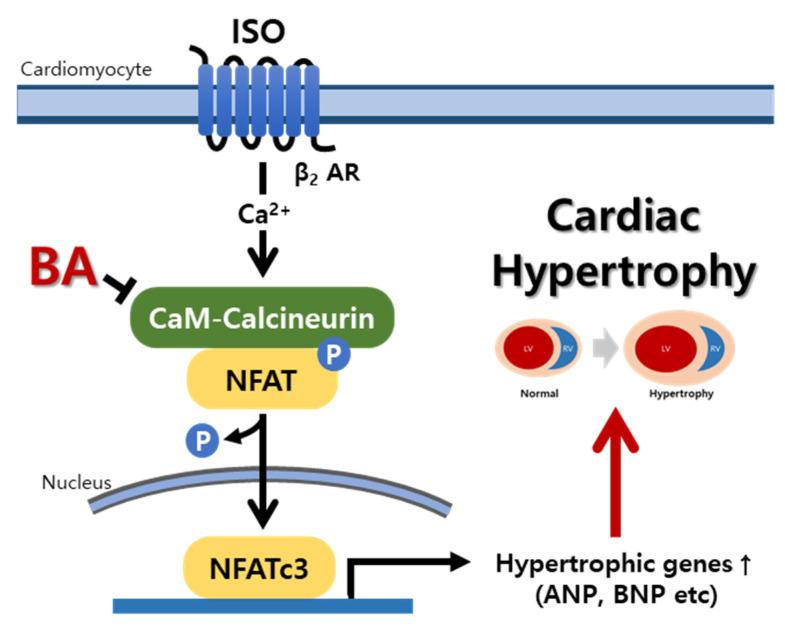
Schematic diagram of the effect of BA on cardio-renal function improvement in cardiac hypertrophy. BA, betulinic acid; ISO, isoprenaline; NFAT, nuclear factor of activated T-cells; ANP. Atrial natriuretic peptide; BNP, brain natriuretic peptide; and β-MHC, beta myosin heavy chain.

**Table 1 nutrients-13-03484-t001:** BA ameliorated cardiac function in isoprenaline-induced LV hypertrophy rat.

	C			Is			Pr			Bl			Bh		
LVEDV (mL)	0.755	±	0.0166	0.89	±	0.066	1.09	±	0.099	0.836	±	0.031	0.80	±	0.086
LVESV (mL)	0.046	±	0.011	0.111	±	0.029	0.184	±	0.031	0.104	±	0.037	0.12	±	0.016
SV (mL)	0.782	±	0.099	0.635	±	0.045 *	0.82	±	0.047 ^#^	0.712	±	0.026 ^#^	0.711	±	0.062 ^#^
LVd Mass	1.28	±	0.071	1.077	±	0.034 *	1.234	±	0.088 ^#^	1.135	±	0.212	1.307	±	0.065 ^##^
%IVS	71.9	±	4.85	71.7	±	5.03	68.7	±	7.45	90.47	±	7.06	55.15	±	2.88
IVSd	1.446	±	0.249	0.868	±	1.99 *	1.808	±	0.150 ^###^	1.723	±	0.154 ^###^	2.068	±	0.087 ^###^
IVSs	3.71	±	0.077	3.29	±	0.14	3.13	±	0.157	3.21	±	0.20	3.35	±	0.104
%LVPW	3.877	±	0.168	2.317	±	0.259 *	3.514	±	0.235 ^##^	4.228	±	0.378 ^#^	4.057	±	0.192
LVPWd	2.23	±	0.093	1.52	±	0.141 **	2.02	±	0.220 ^#^	2.35	±	0.336 ^#^	2.71	±	0.185 ^###^
LVPWs	3.877	±	0.168	2.317	±	0.259 **	3.514	±	0.235 ^##^	4.228	±	0.378 ^##^	4.057	±	0.192 ^###^
LVIDd	7.22	±	0.392	7.21	±	0.164	7.72	±	0.258	7.07	±	0.115	7.16	±	0.245
LVIDs	2.64	±	0.172	2.88	±	0.273	3.79	±	0.304	3.07	±	0.275	3.43	±	0.318

BA, betulinic acid; LV, left ventricle; C, control; Is, hypertrophy rat, isoprenaline (10 mg/kg/day) intraperitoneal injection group; Pr, positive control, Is + propranol (10 mg/kg/day); Bh, Is + betulinic acid (30 mg/kg/day); LVEDV, left ventricular end-diastolic volume; left ventricular end-systolic volume; SV, Stroke Volume; LVd Mass, left ventricular diastolic; %IVS, %interventricular septum; IVSd, Interventricular septal end diastole; IVSs, Interventricular Septum systole; %LVPW, %left ventricular posterior wall; LVPWd, left ventricular posterior wall thickness at end systole; LVPWs, left ventricular posterior wall thickness at end systole; LVIDd, left ventricular end-diastolic diameter; LVIDs, left ventricular end-systolic diameter. Data were expressed as the means ± SD (*n* = 5 for each group). * *p* < 0.05, ** *p* < 0.01 vs. C, ^#^
*p* < 0.05, ^##^
*p* < 0.01, ^###^
*p* < 0.001 vs. Is.

**Table 2 nutrients-13-03484-t002:** BA changed level of serum biomarker in isoprenaline-induced LV hypertrophy rats.

	C			Is			Pr			Bl			Bh		
LDH (U/I)	699.5	±	37.11	1162.5	±	74.16 ***	801.5	±	27.15 ^##^	823.6	±	144.4 ^#^	745.6	±	68.08 ^##^
CK-MB (U/I)	547.2	±	65.9	872.5	±	18.6 **	663.5	±	88.3 ^#^	658.5	±	76.2 ^#^	652.6	±	34.1 ^##^
ALT (U/I)	54.7	±	2.75	72.2	±	6.0 *	54.2	±	0.6 ^#^	53.5	±	3.3 ^#^	56.8	±	2.6 ^#^
AST (U/I)	164.2	±	7.3	186.2	±	9.6	169	±	10.2	168.2	±	17.6	175.5	±	1.4
Albumin (g/dL)	4.07	±	0.19	3.65	±	0.11 *	3.72	±	0.1	3.8	±	0.04	3.8	±	0.13 ^#^
Creatinine (mg/dL)	0.27	±	0.16	0.305	±	0.02	0.295	±	0.0095	0.315	±	0.0025	0.294	±	0.0097

BA, betulinic acid; LV, left ventricle; C, control; Is, hypertrophy rat, isoprenaline (10 mg/kg/day) intraperitoneal injection group; Pr, positive control, Is + isoprenaline (10 mg/kg/day); Bh, Is + betulinic acid (30 mg/kg/day); LDH, lactate dehydrogenase; CK-MB, creatine kinase myocardial band; ALT, alanine aminotransferase; AST, aspartate aminotransferase. Data were expressed as the means ± SD (*n* = 8 for each group). * *p* < 0.05, ** *p* < 0.01, *** *p* < 0.001 vs. C, ^#^
*p*< 0.05, ^##^
*p* < 0.01, vs. Is.

## Data Availability

Not applicable.

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
