# Peer review of "Betulinic Acid Improves Cardiac-Renal Dysfunction Caused by Hypertrophy through Calcineurin-NFATc3 Signaling"

_nutrients, 2021, doi:10.3390/nu13103484_

Round 1
Reviewer 1 Report
Thank you for the opportunity to review the work by Mi Hyeon Hong and colleagues, which I found a very interesting read. I have a few comments: 1. Number of study subjects? 2. The discussion is short and rather technical, and does not include a section concerning limitations, contribution to science and suggestions for future research.
Author Response
We really thank you very much for your invaluable suggestions and comments.
1. Number of study subjects?
→ Thank you for reviewing the submitted manuscript. The number of missing study subjects was added in text, figure legends, and results.
2. The discussion is short and rather technical, and does not include a section concerning limitations, contribution to science and suggestions for future research.
→ As the reviewer suggested, Thanks for reviewing the manuscript submitted. We tried to revise this manuscript grammar overall. Therefore, some fatal grammar errors have been corrected. If we need better proofreading, we can apply to a professional editing company institution anytime.
We really thank Reviewer #1 very much indeed.
Reviewer 2 Report
The manuscript entitled “Betulinic acid ameliorates cardiovascular-renal dysfunction in hypertrophic rats by inhibiting calcineurin-NFATc3 signaling” by Hong et al has investigated the role of Betulinic acid (BA) in isoprenaline (ISO)-induced cardiac hypertrophy rats. The authors observed that the BA-treated rat improved their heart and kidney function in ISO-induced cardiac hypertrophy condition.
Unfortunately, This manuscript has serious issues with experimental design, technical aspects, and data analysis and representation.
The following concerns need to be addressed first before moving further for publication.
- -line no. 23, this sentence is not clear to me as a reviewer, “However, the heart tissue weight ratio body weight was decrease in BA treatment group. ”
- Overall, the abstract is not properly written. The authors should work on the connectivity of sentences. Particularly line no. 22 to 32.
- What is the meaning of β-MHC et al. in line no.55 ?. It needs to be corrected.
- Please expand on the Histological analysis and Measurements of biomarkers in serum (if not in the text, in the figure legend).
- The exact sample size (n) for each experimental group/condition should be mentioned in the manuscript. These are missing in the methods and results.
- Statistical analysis is not performed adequately throughout the results.
- What is the sample size for Figures 1 A and B? The Authors should expand the figure legend.
- In figure 2 A, the scale bar is missing, and in 2 B and C sample size?
- The sample size is missing in table No. 1 and 2
- In figure 3 A, the scale bar is missing.
- In figure 4, sample size and molecular weight( for control and experimental markers) are missing.
- Again sample size is not mentioned in figure 5.
- -line no. 221, remove the “\”
- -line no. 244, please clarify that BA blocks the NFAT expression or phosphorylation of NFAT?
- The conclusion has contradicted the title and results of this manuscript.
Reviewer 3 Report
M&M
- 70: Add “PRO” to this section
- 72: Antibody? Or you bought the recombinant proteins?
- 75: How many rats?
- 100: Please provide the table with exact primary and secondary antibody, dilution and target protein. Then place it according to journal’s requirements in proper M&M section or add to supplementary materials.
- 112: Why did you decide to use the student’s t test instead of ANOVA, after all, you have 5 groups?
- 163: Usually, authors in this particular type of data (pictures) show the Hoechst or DAPI stain to show the nuclei of presented cells. Please add the mentioned staining or explain the lack of those?
- 176: figure 4. A wrong densitometry! Copied from B.
Please mark for each blot molecular weight markers (kDa). Uncropped blots should be placed in supplementary materials.
- 187:
Please calculate creatinine clearance and add to this figure. It is a very important marker for kidney function.
In general, the discussion is too short and does not attempt to compare the results with the available literature. I believe that supposed to be a bit more focused on comparing the results of this work with the available knowledge. The discussion should also include an attempt to explain the phenomena and results presented. In my opinion it should be redrafted.
Round 2
Reviewer 2 Report
The authors made several changes in response to concerns raised in the previous submission. The manuscript has significantly improved.